# Engineering the Vasculature of Stem-Cell-Derived Liver Organoids

**DOI:** 10.3390/biom11070966

**Published:** 2021-06-30

**Authors:** Xv Zhang, Liling Tang, Qian Yi

**Affiliations:** 1Key Laboratory of Biorheological Science and Technology, Ministry of Education, College of Bioengineering, Chongqing University, Chongqing 400044, China; beicang@cqu.edu.cn; 2Department of Physiology, School of Basic Medical Sciences, Southwest Medical University, Luzhou 646000, China

**Keywords:** liver organoids, stem cells, vascularization, maturation, endothelial cells

## Abstract

The vasculature of stem-cell-derived liver organoids can be engineered using methods that recapitulate embryonic liver development. Hepatic organoids with a vascular network offer great application prospects for drug screening, disease modeling, and therapeutics. However, the application of stem cell-derived organoids is hindered by insufficient vascularization and maturation. Here, we review different theories about the origin of hepatic cells and the morphogenesis of hepatic vessels to provide potential approaches for organoid generation. We also review the main protocols for generating vascularized liver organoids from stem cells and consider their potential and limitations in the generation of vascularized liver organoids.

## 1. Introduction

Liver tissues can regenerate after injury but not after chronic fibrosis or cirrhosis. When these diseases progress to the end stage, the only option for treatment is organ transplantation [1]. In one study, the hospital survival rate of patients with acute-on-chronic liver failure (ALF) increased from 16.7% to 62.2% after transplantation [2]. However, the supply of livers for transplantation cannot meet the demand as the number of patients on waiting lists for transplantable livers increases continually [3].

“Organoids” are organ-like tissues self-organized by multiple cell types derived from pluripotent stem cells (PSCs) or isolated organ progenitors. “Organoid therapy” enables the generation of liver organoids using autologous stem cells and exhibits great potential in the treatment of clinically refractory diseases, such as end-stage liver diseases [4]. New techniques permit long-term 3D organoid culture systems for hepatocytes in vitro. The application of growth factors or TNF-α enables in vitro hepatocyte organoid proliferation and hepatocyte repopulation after transplantation [5]. Liver organoids have now been established from tissue-resident stem/progenitor cells [6], embryonic stem cells [7], and induced pluripotent stem cells (iPSCs) [8].

However, major obstacles remain to prevent liver organoids from achieving further maturation. One of the major limitations in the organoid-based generation of functional tissue is that organoids cease to develop a necrotic core after reaching a certain size. As it is not known how to establish an optimal vascular structure at a specific developmental time point during organoid generation, studying vascularization during embryonic development might shed new light on how to design appropriate vascular networks in space and time [9].

Several approaches have been explored to realize better vascularization and maturation of liver organoids. One major breakthrough by Takebe was the generation of vascularized liver buds (LBs) from human iPSCs, which self-organize into 3D iPSC-LBs by interacting with human umbilical vein endothelial cells (HUVECs) and mesenchymal stem cells (MSCs) during organogenesis [8]. However, LBs generated on well arrays were limited in size and unable to form an intact vascular network. Other protocols such as 3D bioprinting, decellularization, and in vivo transplantation have been developed to generate liver organoids with a larger scale and further vascularization.

Herein, we review the major protocols for generating vascularized liver organoids from stem cells. We summarize different theories about the origin of hepatic cells and the morphogenesis of hepatic vessels to provide potential approaches for organoid generation. Finally, we consider the potential and limitations of different protocols in the generation of vascularized liver organoids.

## 2. Liver Vascularization in Embryonic Development

The development of organoids recapitulates committed steps of embryonic cell origin, cell fate specification, cell patterning, organ morphogenesis, and organ growth. To ensure the temporally prescribed occurrence of this development, it is critical to understand the dynamic interaction between tissues and their interpenetrating vascular network. Since we cannot establish an optimal vascular structure at a specific developmental time point during organoid generation, studying vascularization during embryonic development might shed new light on how to design appropriate vascular networks not only in space but also in time [9].

### 2.1. Origin of Hepatic Endothelial Cells

Instead of being built directly from endothelial cells, the bases for hepatic veins, portal veins, and ductus venosus are formed by the anterior portions of the vitelline, umbilical, and posterior cardinal veins, which incorporate into the developing hepatic tissue [10]. In mice, portal veins (PVs) and central veins (CVs) are derived from the fetal vitelline and umbilical veins. During LB delamination and expansion, the umbilical vein and the vitelline vein are surrounded and disrupted by hepatoblasts. Their pre-hepatic and post-hepatic portions form the bases of PVs and CVs [11]. However, only the left umbilical vein is formed during liver organogenesis in humans, and the vitelline veins are not functional [12], which indicates that the left umbilical vein may give rise to the venous system in humans.

More than one source of liver sinusoidal endothelial cells (LSECs) has been studied. Sugiyama demonstrated that LSECs might originate from the omphalomesenteric veins, posterior cardinal veins, or common cardinal veins. Flk-1 and PECAM-1 are vascular endothelial growth factor (VEGF) receptors expressed by ECs close to the liver diverticulum at E9.5 (Figure 1a). Immunohistochemistry analysis of Flk-1 and PECAM-1 expression in serial sections of the liver primordium revealed that all of the ECs around the liver diverticulum were well connected with each other and also with large vessels around the liver primordium, including the omphalomesenteric veins and common or posterior cardinal veins, which suggested that ECs of these large vessels may invade the liver diverticulum and then give rise to hepatic sinusoids [13]. A recent study of mice revealed that the sinus venosus endocardium, which flanks the liver at the budding stage, contributes to more than one-third of the ECs of the sinusoids, the CVs, and the PVs (Figure 1b). The sinus venosus-derived cells are attracted by VEGF secreted by hepatoblasts [14]. In parallel, endothelial cell progenitors were found in the endoderm by lineage tracing using Foxa2:T2AiCre or Foxa2:CreER mice. FOXA2 is an endoderm marker, and to track the fate of FOXA2-expressing cells, Goldman et al. crossed the heterozygous Foxa2-iCre mouse with the homozygous reporter-enhanced yellow fluorescent protein (YFP) mouse. YFP can be ubiquitously expressed under the Rosa26 promoter only after Cre recombinase excises the STOP cassette flanked by LoxP sites. Flow analyses for the co-expression of YFP with the endothelial markers CD31, KDR, and CD144 in dissociated E13.5 fetal livers showed that about 15% of the CD144+KDR+ EC population within E13.5 fetal livers was YFP+, which suggested the percentage of ECs derived from the endoderm was around 15% [15]. In addition, cell population analysis of the liver revealed that hepatic ECs might originate from hepatic progenitor cells. Examination of a wild-type embryonic liver at E10.5 and E14.5 showed the appearance of Flk1/Gata4 double-positive ECs in tissue surrounded by hepatocytes that only expressed Gata4. Cell population analysis of E12.5 embryonic livers by flow cytometry showed that the population of VE-cadherin positive ECs is co-expressed Gata4, which suggested that both ECs and hepatocytes are derived from Gata4-positive progenitor cells [16].

### 2.2. Morphogenesis of Veins, Arteries, and Sinusoids

Vascularization during liver morphogenesis is determined by the venous channels other than the arterial supply [9]. The PV is the first branched structure, providing a frame for the development of the biliary network, the branching of the hepatic artery, and the expansion of hematopoietic stem cells. The portal mesenchyme instructs ductal plate development, and biliary cells subsequently induce the morphogenesis of the hepatic artery [17].

The development of the hepatic artery depends on VEGF from cholangiocytes, angiopoietin from hepatoblasts, and VEGFR-2 and Tie-2 receptors from precursors of ECs and mural cells (MCs), respectively. At the ductal plate stage, VEGF activates VEGFR-2-positive EC precursors and recruits them to the portal mesenchyme near the ductal plate, where they express VEGFR-1 in the form of small cell clumps. Meanwhile, TIE-2-positive portal myofibroblasts are recruited by ANG-1 as MC precursors in the portal space. During the migratory stage, VEGFR-1-positive ECs and TIE-2-positive MCs assemble into an immature hepatic artery. At the time of the bile duct stage, ECs acquire TIE-2 and VEGFR-1, making it easier for VEGF and Ang-1 to induce lumen formation and progressive vascular enlargement for the maturation of the hepatic artery [18].

Within hepatic sinusoids, LSECs are highly specialized. Notably, they lack a basement membrane and have open fenestrae clustered in sieve plates. These structures enable the free transfer of fluid, ions, nutrients, and metabolites, and also direct contact between the parenchymal liver cells (hepatocytes) and non-parenchymal cells such as hepatic stellate cells [19]. These characteristics of LSECs in mature sinusoids are progressively acquired through a three-phase process: the earliest non-fenestrated LSECs are surrounded by a laminin-rich basement membrane and express non-specific endothelial markers, such as CD31, CD34, and IF10. Then, with the changes in the perisinusoidal extracellular matrix (ECM), the LSECs are fenestrated. CD31 and CD34 are decreased, while the expression of more specific adult sinusoid markers, such as CD4, CD32, and ICAM-1, is increased. These changes are the first signs of stereotyped differentiation of LSECs. Finally, the large fenestraes on sinusoids progressively disappear, and the sinusoids with small fenestrae markedly increase and become zonate at birth [17]. The mechanisms of LSEC differentiation in the fetal liver remain unclear. Some studies support that VEGF and Wnt signaling have important functions in LSEC proliferation and differentiation. By activating the canonical β-catenin pathway, Wnt2 can increase LSEC proliferation [20]. Loss of VEGF in hepatic cells indicates that VEGF may react to the number and capillarization of LSECs [21]. Recent research in mice demonstrated that the transcription factor GATA4 controls LSEC specification and functions. LSEC-restricted deletion of Gata4 resulted in the transformation of discontinuous liver sinusoids into continuous capillaries. Correspondingly, the ectopic expression of Gata4 mediated the downregulation of continuous EC-related transcripts and the upregulation of LSEC-related genes in continuously cultured ECs [22].

## 3. Vascularization and Maturation of Liver Organoids

### 3.1. Self-Organization in Well Arrays

In 2013, Takebe et al. generated the first vascularized LB by coculturing human-iPSC-derived hepatic endoderm (hiPSC-HE) cells, HUVECs, and bone MSCs on Matrigel, which recapitulated early organogenesis. These cells were plated in two-dimensional conditions but self-organized into macroscopically visible three-dimensional cell clusters by an intrinsic organizing capacity at 48  h after seeding. After 6 days of culture, there were visible CD31+ vascular networks. Compared to LBs generated with hepatocytes derived from hiPSCs by other protocols, the structural configuration of these hiPSC-LBs was closer to the adult human liver despite its fetal transcription profile. Forty-eight hours after being transplanted in a cranial window mouse model, human blood vessels within the LBs connected to the host vessels at the edge of the transplant. Forty-five days after transplantation, the presence of albumin-positive cells (detected by flow cytometry) and secreted human albumin (detected by ELISA) in the serum of mice revealed further maturation of the transplanted hiPSC-LBs [8]. In 2017, Takebe et al. developed a highly reproducible method to generate LBs entirely from feeder-free hiPSCs using the microwell-array-based approach. For the mass production of LBs, they developed an omni-well-array culture platform, which could produce homogeneous and miniaturized LBs on a large, clinically relevant scale (>108). Although no vascular network was formed in vitro, many angiogenesis hallmark gene sets were observed in d-8 LBs, and functional blood vessels formed 2 days after the LBs were transplanted into immunodeficient mice [23].

Endothelial cells from other sources have also been applied to vascularize LBs. Recently, Pettinato et al. staggered human adipose microvascular endothelial cells (HAMECs) with hiPSCs on micro-well arrays. Tests of persistent albumin secretion post-transplantation in animal models demonstrated that clusters with HAMECs showed consistently better maturation than the hiPSCs-only group [24]. In 2019, Li et al. successfully assembled human LBs containing LSECs and hepatic stellate cells (HSCs). They generated LBs from naïve MSCs, MSC-derived hepatocytes, and HSC- and LSEC-like cells in an optimized 10:3:1:1 ratio in a 24-well plate. Within 72 h, the cell mixture self-assembled into human LBs in vitro, and characteristics similar to early stage murine LBs were exhibited. After being transplanted into a murine model of ALF, the LBs effectively reduced the likelihood of animal death and triggered better hepatic ameliorative effects than splenic transplantation of naïve MSCs or human hepatocytes. Furthermore, transplanted human LBs underwent further maturation during injury alleviation and subsequently exhibited gene expression profile characteristics similar to those of adult livers [25]. In 2020, Kiryu et al. generated liver-specific vascularized hepatobiliary organoids by combining mouse liver progenitor cells (LPCs) with mouse LSECs. During a 7-day culture, the organoids developed clusters of polygonal hepatocyte-like cells and biliary ducts. Two weeks after being transplanted into vascularized chambers of Fah-/-/Rag2-/-/Il2rg-/- mice, the organoids generated only from LPCs presented minimal surviving LPCs in chambers, but those generated with both LPCs and LSECs presented robust hepatobiliary ductular tissue. These results confirmed that the incorporation of LSECs with LPCs into organoids significantly improved the differentiation and survival of organoids after transplantation [26].

To better direct the vascularization and maturation of liver organoids in vitro, Velazquez et al. applied comprehensive analysis and gene regulatory networks (GRNs) engineering to generate PSC-derived multilineage human liver organoids. In their previous work, they engineered the process by which hiPSCs self-organized into a fetal liver organoid (FeLO) by transient lentiviral expression of GATA6. The FeLO presented a limited vascular network and immature hepatic characteristics [27]. In order to improve the maturation of organoids, they devised a framework using FeLO as a testbed to rationally direct FeLO morphogenesis with the assistance of unbiased computational analysis and GRNs reprogramming. By overexpressing PROX1 and ATF5, combined with targeted CRISPR-based transcriptional activation of endogenous CYP3A4 in *GATA6*-engineered hiPSCs, they generated a tissue called designer liver organoid (DesLO) that was rationally invented and genetically engineered. In DesLO, many key genes related to angiogenesis were induced, including VEGFA, placental growth factor (PGF), and kinase insert domain receptor (KDR), which generated a vast, interconnected vascular network in DesLO revealed by image analysis of vasculature. Compared to FeLO, the total length of the vessel, the percentage area of the vessel, and the number of vascular junctions were increased in DesLO. In addition, vessel metrics showed that the DesLO cultures remained stable while FeLO decreased from day 14 to 17, indicating the better stability of the vascular network in DesLO [28].

In summary, different sources of ECs combined with stem-cell-derived hepatic endoderm cells or LPCs have been used to generate LBs. The use of endothelial cells is indispensable to the formation of 3D transplantable tissues [8] and can improve the vascularization and maturity of liver buds. However, self-assembled LBs in vitro are limited in size (from 250 μm [26] to 4 mm [25] in diameter), and they also tend to have immature hepatic characteristics and limited vascular networks.

### 3.2. Three-Dimensional Bioprinting

Filament deposition fabrication (FFF) and fused deposition modeling (FDM) are the methods most frequently used to generate large-scale tissue constructs with more complete vascular networks. FDM has been used to vascularize tissue in vitro. In a pre-programmed and controlled manner, a filament-like hydrogel that contains ECs interleaved or combined with other cell types is extruded and deposited to form a specific pattern [29].

In 2020, Kang et al. created an array of hepatic lobules (~1 mm) with a preset extrusion bioprinting technique. A high cell density of HepG2/C3A and ECs was embedded on the lobule structure with a central vein using a preset cartridge. ECs covered the outside of the construct, forming a lumen in which ECs were lined and interconnected between the exterior and the lumen to maintain cell survival in the lobule structure [30] (Figure 2).

By fusing hundreds of LB-like spheroids with a 3D bioprinter, Yusuke et al. succeeded in developing a scaffold-free method for the rapid fabrication of scalable liver-like tissue. They mixed early lineage hepatocytes, HUVECs, and MSCs to generate LBs, and subsequently used a needle-array system to fix the LBs into 3D tissue to fabricate elaborate geometries instead of a scaffold, which could avoid problems such as infection, immune response, and the degradation of exogenous substances. The needle-array system could also prevent an ischemic environment in vitro by executing culture circulation immediately after bioprinting. The human-liver-like tissue self-organized in vitro, and after being transplanted on a rat liver, it survived within 100 µm [31].

In 2021, Yang et al. generated hepatorganoids by 3D bioprinting of HepaRG cells without ECs, which became vascularized and functional in vivo after transplantation. HepaRG cells differentiated into hepatocytes and generated 3D bioprinted hepatorganoids (3DP-HOs) in vitro. Then, the 3DP-HOs formed functional vascular systems. The survival of Fah-/-Rag2-/- (F/R) mice with liver injury was significantly improved by transplantation of 3DP-HOs, which suggested that the 3DP-HOs from HepaRG cells could rescue liver injury [32].

Bioprinting allows for the fabrication of engineered tissues with spatial organization of heterogeneous cells and ECM to form functional organs, enabling more accurate modeling of native tissue organization, especially the integration of vascular networks within the tissues [28]. However, the printing resolution is limited. For example, the printing resolution is 100 µm in extrusion-based bioprinting [33], but the diameter of a typical capillary is about 10 µm. Furthermore, the liver is highly vascularized, so a vascular network is indispensable for its function, meaning that the interconnection of microchannels during bioprinting is essential [34].

### 3.3. Decellularization

In recent years, decellularized matrices derived from donor livers have been applied as scaffolds to fabricate transplantable liver-l0069ke structures with hepatic functions. The liver’s general structure and vascular architecture can be preserved in a decellularized liver matrix [35]. Stem cells and ECs are perfused through the vascular network, and the scaffolds are refilled by engrafting them into their presumed natural locations in the liver [36].

In 2010, Baptista et al. successfully fabricated 3D scaffolds with intact vascular networks based on decellularization. The liver tissue from different species (mice, rats, ferrets, rabbits, and pigs) was decellularized by detergent perfusion to remove the cellular components selectively while preserving the intact vascular networks and the ECM component. The vascular network had one central inlet accessible to the whole vasculature, and a capillary-like network was branched from this inlet and then reunited into one central outlet. Human fetal liver cells and ECs were perfused through the vascular network to repopulate the scaffold by engrafting into their putative locations. When the typical endothelial, hepatic, and biliary epithelial markers were displayed, they indicated that hepatic tissue had been created in vitro [36].

Liver organoids generated from a decellularized matrix can be a powerful tool for regenerative medicine. However, since an acellular ECM is potently thrombogenic, an insufficiently endothelialized construct will likely cause blood clots, even when using standard anticoagulant therapy. To prevent thrombosis, it is crucial to reestablishing a sufficiently re-endothelialized patent vasculature [37].

To maximize endothelial cell coverage of the vessel walls, Ko et al. conjugated anti-endothelial cell antibodies to reestablish a vascular network within decellularized liver scaffolds. This protocol achieved uniform EC coverage throughout the vascular networks, including the capillary bed, which in vitro greatly reduced platelet adhesion during blood perfusion. After transplantation into recipient pigs, the re-endothelialized scaffold could withstand the physiological blood flow and maintain its structure for 24 h. This was the first vascularized bioengineered liver with a similar clinical size that could be transplanted and maintained in vivo [38].

Another antithrombotic coating reagent used to induce the attachment of ECs on vascular walls is the heparin-gelatin (HG) mixture. Hussein et al. coated PVs and hepatic artery walls with the HG mixture and re-endothelialized the scaffold with ECs. The re-endothelialized scaffold was maintained in a bioreactor with blood flow for 10 days. ECs efficiently covered the vascular network and maintained function and proliferation. After 24 h of blood perfusion, the HG-precoated scaffolds were determined by thrombogenicity evaluation. No thrombosis was detected, which demonstrated that the vascular tree was efficiently endothelialized. After the coculture of HepG2 cells and ECs, HepG2 cells presented a higher function in HG-precoated scaffolds compared to uncoated scaffolds [39].

Decellularized matrices derived from donor livers can generate whole-liver scaffolds of clinically relevant size for clinical applications, such as treating patients with end-stage liver diseases [37]. However, the current fabrication of solid organs fails to meet the standard of clinical use in several aspects, including immune responses after transplantation, preservation of functional ECM structure, and long-term functional integration.

### 3.4. In Vivo Transplantation

Organoids with fully functional vascularization have only been generated through transplantation into host animals, where ectopic implants were seen to be integrated with native vasculature. Most often, in vitro generation of organoids required engraftment for further vascularization.

In 2014, Takebe et al. generated 4-day-old iPSC-LBs on a pre-solidified matrix and transplanted them into immunodeficient mice at various ectopic sites, including under a cranial window, under the kidney capsule, onto the distal mesentery, and onto the proximal mesentery. Live imaging revealed functional blood perfusion into the preformed human vascular networks within LBs [40]. Li et al. successfully prevented death in mice using self-assembled human LBs through mesenteric transplantation, which presented better hepatic protein production and injury amelioration. This method is more clinically relevant and less invasive than orthotopic transplantation methods because the blood flow from PVs is not affected by mesenteric transplantation, which is crucial to improve hepatic functions [26].

Mavila et al. generated a functional tissue-engineered liver (TELi) through transplantation with the liver organoid unit (LOU), prepared from 2-week-old ActinGFP mice or human liver and implanted it into host bodies. At the time of harvest, there were visible branched capillaries on the surface of TELi. Vascular structures similar to the native liver were demonstrated by the presence of α-SMA-expressing portal fibroblast cells and CD31− expressing ECs [41].

The transplantation of LBs in mice can lead to further vascularization and maturation. However, in vivo vascularization of organoids is expensive, resource-intensive, complex in process, and likely to cause immune rejection; it also lacks control and scalability during the tissue-development processes.

Each of the methods mentioned above has its advantages and disadvantages. LBs self-organized in well arrays can be easily cultured without sophisticated equipment. Still, their size in vitro is limited (from 250 μm [26] to 4 mm [25] in diameter), and they also tend to have immature hepatic characteristics and limited vascular networks. Three-dimensional bioprinting is capable of generating large-scale tissue constructs with more complete vascular networks. However, the printing resolution is limited to 100 µm in extrusion-based bioprinting [33], whereas the diameter of a typical capillary is about 10 µm, which makes it difficult to construct microchannels with interconnection. Decellularization can preserve intact vascular architecture in a decellularized liver matrix [36], generating whole-liver scaffolds of clinically relevant size. However, organoids generated by decellularization fail to meet the standard of clinical use in regard to immune responses after transplantation, preservation of functional ECM structure, and long-term functional integration. Transplantation into host animals is the only way through which organoids can achieve fully functional vascularization, but in vivo vascularization of organoids is resource-intensive and complex in process, lacks control during development, and is likely to cause immune rejection after implantation. All in all, these methods are precluded from further application in regenerative medicine by their limitations.

## 4. Conclusions

To achieve better vascularization and maturation, different strategies have been developed for the generation of liver organoids (Table 1). Currently, organoids are useful for studying liver development, modeling liver diseases, and screening for hepatotoxicity. Decellularized matrices derived from donor livers can generate human-sized whole-liver scaffolds with intact vascular networks. Transplantation to host animals can lead to a certain degree of vasculature maturation. However, the generation of a fully functional liver with a specialized vasculature and bile duct system has not been accomplished. Moreover, before liver organoids are applied in regenerative medicine, urgent safety issues, such as immunological rejection, must be addressed.

## Figures and Tables

**Figure 1 biomolecules-11-00966-f001:**
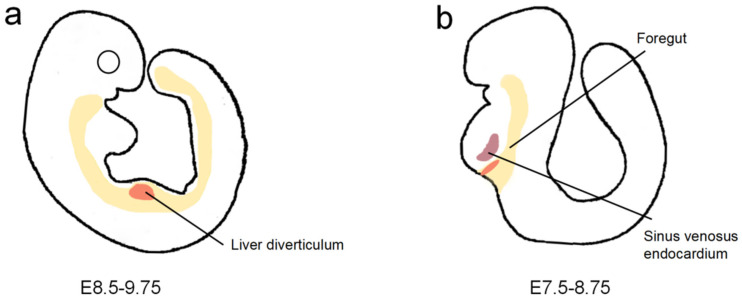
The origin of hepatic endothelial cells. (**a**) At E9.5, the liver diverticulum is surrounded by the omphalomesenteric veins, posterior cardinal veins, or common cardinal veins. (**b**) At E8.5, the liver at budding stage is located in the portion of the ventral foregut endoderm, which is adjacent to the sinus venosus endocardium.

**Figure 2 biomolecules-11-00966-f002:**
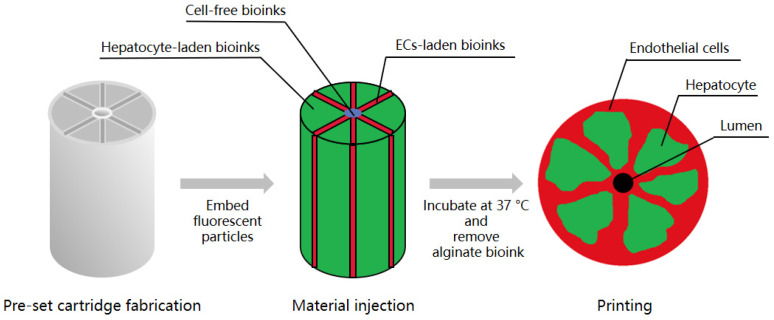
Overview of the protocol for the generation of a hepatic lobule array. Kang et al. fabricated a preset cartridge and injected hepatocyte-laden bioinks, ECs-laden bioinks, and cell-free bioinks into it to imbed HepG2/C3A and ECs. Then, they removed the alginate bioink to form a lumen. After incubation at 37 °C, ECs covered the outside of the construct, forming a lumen in which ECs were lined and interconnected between the exterior and the lumen.

**Table 1 biomolecules-11-00966-t001:** Summary of the current published protocols for the vascularization of liver organoids.

Method	Material	Vascularization Strategy	References
Self-organization in well arrays	iPSC-HE cells, HUVECs, and BMSCs	Coculture of HUVEC, MSCs, and hPSC-derived hepatic progenitors. MSC-driven condensation on Matrigel	[8]
	Feeder-free human iPSCs	Coculture of iEC, iMSCs, and hiPSC-derived hepatic progenitors iMSC-driven condensation on Matrigel	[23]
	HAMECs and hiPSCs	EB differentiation of hepatic cells in the presence of HAMEC	[24]
	MSCs, MSC-derived hepatocytes, and HSC- and LSEC-like cells	Coculture of MSCs, MSC-derived hepatocytes, and HSC- and LSEC-like cells MSC-driven condensation on Matrigel	[25]
	LPCs and LSECs	Coculture of LPCs and LSECs	[26]
	Lentivirus vector, hiPSCs, and primary human hepatocytes	Engineering of GRN by lentiviral transduction	[28]
Three-dimensional bioprinting	HepG2/C3A cells, EA.hy 926 cells, fabricated alginate solution for cell-laden bioinks and sacrificial materials, lyophilized Atelocollagen, and gelatin powder to form scaffold	Embedding of ECs on a lobule structure with microchannel built by sacrificial material to form an endothelium-lined lumen	[30]
	Liver-bud-like spheroids generated by mature hepatocytes, HUVECs, and MSCs	Coculture of mature hepatocytes, HUVECs, and MSCs	[31]
	HepaRG cells, sodium alginate solution, and gelatin solution for bioink	In vivo perfusion after transplantation	[32]
Decellularization	Liver tissue from mice, rats, ferrets, rabbits, and pigs, 1% Triton-X 100 with 0.1% ammonium hydroxide for decellularization, HUVECs, hFLCs	Seeding ECs on the decellularized liver with a vascular network	[36]
	Porcine liver harvested from 5 to 8 kg piglets, 1% Triton X-100 and 0.1% ammonium hydroxide in distilled water for decellularization, rat anti-mouse CD31 antibody to improve re-endothelialization, and vascular endothelial cells expressing GFP protein (MS1)	Seeding ECs on the decellularized liver with a vascular network	[38]
	Porcine livers collected from adult pigs (40–50 kg), 1% Triton X-100 and 0.1% ammonium hydroxide in distilled water for decellularization, HG mixture to improve re-endothelialization, human EA.hy926 endothelial cells, hepatic carcinoma cells (HepG2 cells)	Seeding ECs on the decellularized liver with a vascular network	[39]
In vivo transplantation	LBs generated from iPSC-HE cells, HUVECs and BMSCs, immunodeficient mice as hosts	Blood perfusion in host body after transplantation	[40]
	LBs generated from LPCs and LSECs, using Fah-/-/Rag2-/-/Il2rg-/- mice as hosts	Blood perfusion in host body after transplantation	[26]
	LOU prepared from human liver or 2-week-old ActinGFP mice, NOD/SCID γ host mice as hosts	Blood perfusion in host body after transplantation	[41]

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
