# Peer review of "Engineering the Vasculature of Stem-Cell-Derived Liver Organoids"

_biomolecules, 2021, doi:10.3390/biom11070966_

Round 1
Reviewer 1 Report
General comment:
The review “Vascularization of stem cell-derived liver organoids” is focusing on recent advantages on liver organoids generation centering on the importance of vascularization using primary endothelial cells or differentiated from human iPSCs. It is very interesting the focusing on embryonic development of endothelial cells to better understand the process and to improve the development of vascularized organoids. Therefore, some minus revisions are required.
Revision:
- It can be useful to add a separate paragraph on advantages and disadvantages of the different methods to vascularize organoids. These information are already listed through the manuscript, but to group them in a separate section can help the understanding.
- The numbering of the figures is not correct. There are two figure 1. Moreover, in the figures it is recommended of not adding references. So please rephrase the figure legend.
- Reference number 23 is not correct in the text. It is not Giuseppe et a., but Pettinato et al.
Author Response
Response to Reviewer 1:
Comment 1: It can be useful to add a separate paragraph on advantages and disadvantages of the different methods to vascularize organoids. These information are already listed through the manuscript, but to group them in a separate section can help the understanding.
Answer: We have added a paragraph on advantages and disadvantages of the different methods to vascularize organoids in the end of section 3.
Comment 2: The numbering of the figures is not correct. There are two figure 1. Moreover, in the figures it is recommended of not adding references. So please rephrase the figure legend.
Reference number 23 is not correct in the text. It is not Giuseppe et al., but Pettinato et al..
Answer: We have corrected the numbering of the figures and reference number 23, we also rephrased the figure legend.
Reviewer 2 Report
Vascularization of stem cell-derived liver organoids
Xv Zhang, LilingTang and QianYi
The review addresses a very interesting aspect of liver organoids, the (lack of) vascularisation. The title seems to suggest that this is possible and the abstract also states “Liver organoids generated from human stem cells can be vascularized using methods that recapitulate the embryonic liver development “ as if “we” can already do this. The review shows that this is not the case so the title and the abstract should not suggest this. The title should reflect that vascularization is not working yet.
Statements on page 1 “Furthermore, cirrhosis is irreversible and able to cause hepatocellular 22
carcinoma.” This is not true. Cirrhosis is reversible as has been shown by the recent HepC treatments. As long as one can take away the causing agent, and the liver is not at end stage liver disease, cirrhosis is reversible.
The grammar needs to be checked by a native speaker. It seems that many sentences have been taken a bit out of context. The sentence “New techniques permit long-term expansion of hepatocytes in vitro, which provides sufficient cells for organoid generation” , but the paper cited show that the hepatocytes are being expanded as organoid cultures.. The next sentence starts with And, to follow with a list of source from which organoids have been made (not for multiple species), but then citing a review that states this..Why not cite the original papers or state that is is reviewed in.
The text is just not easy to read.. There are too many errors such as “However, there remains major obstacles”, and “Therefore, a major challenge in the field remains to be achieving further vascularization”, “A recent studies in mice have revealed”, “the population of VE-cadherin positive ECs is co-expressed Gata4,” “Within hepatic sinusoids, LSECs are highly specialized, they are lack of basement membrane and have open fenestrae clustered in sieve plates” etc. and many others.. I will not highlight these anymore, but a native speaker should read this text cause it is very frustrating to go through the entire text and find these kind of sentences.
“Therefore, an important issue of organoid vascularization is to design appropriate vascular network, not only in space but also in time [7], which may be answered by the studyof vascularization during embryonic development.” One states that there may be an answer, but there is no question raised. There was an issue.. studying vascularization during development might shed new light on how to design appropriate vascular networks…
During their description of the origin of hepatic endothelisal cells one should design and show a figure illustrating this. For most people this is difficult to imagine.. while for instance Figure 1 can be ommitted because it is a bad representation of what is in the Takebe paper.
Not clear what is meant by “Immunohistochemistry analysis for these receptors revealed the intact connection between all ECs and ECs, ECs and vessels around the liver diverticulum, which suggested that ECs of these large vessels may invade the liver diverticulum and then give rise to hepatic sinusoids”
“Using Foxa2:T2AiCre mice,”, it is not clear why these mice were used and what was traced?
On page three the authors write a paragraph about “The development of artery….” and cite a manuscript form the Roskams lab which is purely descriptive (and used an Hnf6KO mice), but these conclusions that VEGF and Ang-1 act synergistically cannot really be drawn from this study.
The statement “These structures probably play a role in the transfer of substrates between hepatocytes and bloodstream [18].” Is really an understatement, The function of LSECs has been well described by many groups.. The LSEC differentiation aftwards and the reference from which they took the phrases could also be referred to as reviewed in 16.. instead of just 16.
Section 3.
The description of the Takebe papers does not add much to the review.. Yes, there were cd31 positive cells in their LBs, and indeed when tranplanted there was a clear connection of HUVECs with the mouse vasculature.. How this happened was unclear.. And the highly reproduceible method for LB did not show vasculature…
Author Response
Response to Reviewer 2:
Comment 1: The title seems to suggest that this is possible and the abstract also states “Liver organoids generated from human stem cells can be vascularized using methods that recapitulate the embryonic liver development “ as if “we” can already do this. The review shows that this is not the case so the title and the abstract should not suggest this. The title should reflect that vascularization is not working yet.
Answer: We have modified the title and abstract. We have replaced “Vasularization” in the title with “Engineering the vasculature” in the hope of avoiding misleading. We also have changed “Liver organoids generated from human stem cells can be vascularized” in the abstract to “Vasculature of stem cell-derived liver organoids can be engineered”.
Comment 2: Statements on page 1 “Furthermore, cirrhosis is irreversible and able to cause hepatocellular 22carcinoma.” This is not true. Cirrhosis is reversible as has been shown by the recent HepC treatments. As long as one can take away the causing agent, and the liver is not at end stage liver disease, cirrhosis is reversible.
Answer: We have deleted the statement and changed the sentence to “When these diseases progress to the end-stage, the only option for treatment is organ transplantation”.
Comment 3: The grammar needs to be checked by a native speaker. It seems that many sentences have been taken a bit out of context. The sentence “New techniques permit long-term expansion of hepatocytes in vitro, which provides sufficient cells for organoid generation” , but the paper cited show that the hepatocytes are being expanded as organoid cultures.. The next sentence starts with And, to follow with a list of source from which organoids have been made (not for multiple species), but then citing a review that states this..Why not cite the original papers or state that is is reviewed in.
Answer: We have commissioned LetPub, a professional language polishing company, to polish this article. And we have corrected the information of this sentence and changed it to “New techniques permit long-term 3D organoid culture system for hepatocyte in vitro”. We also have added references for each source mentioned in the next sentence and deleted “for multiple species”.
Comment 4:The text is just not easy to read.. There are too many errors such as “However, there remains major obstacles”, and “Therefore, a major challenge in the field remains to be achieving further vascularization”, “A recent studies in mice have revealed”, “the population of VE-cadherin positive ECs is co-expressed Gata4,” “Within hepatic sinusoids, LSECs are highly specialized, they are lack of basement membrane and have open fenestrae clustered in sieve plates” etc. and many others.. I will not highlight these anymore, but a native speaker should read this text cause it is very frustrating to go through the entire text and find these kind of sentences.
Answer: Thank you very much for your useful comments and suggestions. We have commissioned LetPub, a professional language polishing company, to polish this article. Errors and grammar problems throughout this article have been corrected, hope to make it easier to read.
Comment 5: “Therefore, an important issue of organoid vascularization is to design appropriate vascular network, not only in space but also in time [7], which may be answered by the studyof vascularization during embryonic development.” One states that there may be an answer, but there is no question raised. There was an issue.. studying vascularization during development might shed new light on how to design appropriate vascular networks…
Answer: We have modified the sentence and the question we wanted to raise is how to establish an optimal vascular structure at a specific developmental time point during organoid generation.
Comment 6: During their description of the origin of hepatic endothelisal cells one should design and show a figure illustrating this. For most people this is difficult to imagine.. while for instance Figure 1 can be ommitted because it is a bad representation of what is in the Takebe paper.
Answer: We have deleted Figure 1 and added a new figure (Figure 1) to illustrate the origin of LSECs at the end of section 2.1.
Comment 7: Not clear what is meant by “Immunohistochemistry analysis for these receptors revealed the intact connection between all ECs and ECs, ECs and vessels around the liver diverticulum, which suggested that ECs of these large vessels may invade the liver diverticulum and then give rise to hepatic sinusoids”.
Answer: We have corrected this sentence and changed it to “Immunohistochemistry analysis of Flk-1 and PECAM-1 expression in serial sections of the liver primordium revealed that all of the ECs around the liver diverticulum were well connected with each other and also with large vessels around the liver primordium, including the omphalomesenteric veins, and common or posterior cardinal veins”.
Comment 8: “Using Foxa2:T2AiCre mice,”, it is not clear why these mice were used and what was traced?
Answer: We have described the experiment in more detail. We have described the role of different genes in Foxa2:CreER mice to explain why these mice were used and added that the fate of FOXA2-expressing cells was tracked.
Comment 9: On page three the authors write a paragraph about “The development of artery….” and cite a manuscript form the Roskams lab which is purely descriptive (and used an Hnf6KO mice), but these conclusions that VEGF and Ang-1 act synergistically cannot really be drawn from this study.
Answer: We have deleted the statement that VEGF and Ang-1 act synergistically and corrected the conclusion.
Comment 10: The statement “These structures probably play a role in the transfer of substrates between hepatocytes and bloodstream [18].” Is really an understatement, The function of LSECs has been well described by many groups.. The LSEC differentiation aftwards and the reference from which they took the phrases could also be referred to as reviewed in 16.. instead of just 16..
Answer: We have searched another paper and revised this statement, hope to describe the function of LSECs better. We also changed reference number 16 to (reviewed in Ref. [17]).
Comment 11: The description of the Takebe papers does not add much to the review.. Yes, there were cd31 positive cells in their LBs, and indeed when tranplanted there was a clear connection of HUVECs with the mouse vasculature.. How this happened was unclear.. And the highly reproducible method for LB did not show vasculature…
Answer: We have added details about how the cells self-organized and how human blood vessels within the LBs connected host vessels to the description of the Takebe paper in the first paragraph of section 3.1, hope to describe their paper in 2013 better. We have also added more details about the result of their research of the highly reproducible method in the first paragraph of section 3.1, which may prove their work can be used to mass produce LBs with vascular.
Round 2
Reviewer 2 Report
The authors have clearly followed the suggested of the first review round. However, still many typo's and english grammar mistakes are in this review.
Normally LetPub gives an edited word document with all track changes. Here I see only the track changes in the sections where the reviewers have requested the changes.... but sentences like "These changes are the first stereotyped differentiation of LSECs." remain.. Which do not mean much.. and might not have been picked up by the editing service.
Author Response
Dear Editor and Reviewers,
Thank you very much for your useful comments and suggestions on our manuscript. We have modified the manuscript accordingly as follow:
Response to Reviewer 2:
Comment 1: The authors have clearly followed the suggested of the first review round. However, still many typo's and english grammar mistakes are in this review.
Normally LetPub gives an edited word document with all track changes. Here I see only the track changes in the sections where the reviewers have requested the changes.... but sentences like "These changes are the first stereotyped differentiation of LSECs." remain.. Which do not mean much.. and might not have been picked up by the editing service.
Answer: Thank you very much for your useful comments and suggestions. We have commissioned LetPub to polish this article again. Errors and grammar problems that have not been corrected have been revised. We have retained LetPub's track changes this time and checked whether there were any unmodified errors.
The manuscript has been resubmitted to your journal. We look forward to your positive response.
Sincerely,
Liling Tang